# Uncontrolled hypertension among adult hypertensive patients in Addis Ababa public hospitals: A cross-sectional study of prevalence and associated factors

**Asmamaw Deguale Worku**[1,2]*, **Asinake Wudu Gessese**[3]

1 Department of Water and Health, Ethiopian Institute of Water Resources, Addis Ababa University, Addis Ababa, Ethiopia, 2 Department of Public Health Emergency Management, Addis Ababa Health Bureau, Addis Ababa, Ethiopia, 3 Department of Public Health, Yekatit 12 Hospital Medical College, Addis Ababa, Ethiopia

* asmamawdeguale16@gmail.com

**Data Availability Statement:** All necessary materials used and/or analyzed during the current study were included in the manuscript.

## Abstract

### Background

In 2019, 77% of women and 82% of men with hypertension had uncontrolled hypertension worldwide. Uncontrolled hypertension can cause stroke, myocardial infarction, heart failure, renal failure, dementia, blindness, and death. However, most of the studies used the previous seventh joint national committee classification to classify hypertensive patients as either controlled or uncontrolled. This study aimed to assess the prevalence and associated factors of uncontrolled hypertension among adult hypertensive patients at public hospitals in Addis Ababa, Ethiopia.

### Methods

From April 12 to May 12, 2024, three public hospitals in Addis Ababa employed a hospital-based cross-sectional study design with 408 hypertensive patients. Systematic random sampling was used to select the study participants. We used a structured interview questionnaire and chart review and took physical measurements. Data were entered into Epidata and analyzed using the statistical package for social science version 25. A logistic regression model was used to identify factors associated with uncontrolled hypertension at a P-value < 0.05 with a 95% confidence interval.

### Results

The prevalence of uncontrolled hypertension among hypertensive patients at public hospitals in Addis Ababa was 66.2% (95% CI: 61.6%, 70.8%). After adjusted analysis, age ≥ 60 years (AOR = 2.88, 95% CI: 1.37, 6.04), the presence of comorbidities (AOR = 2.21, 95% CI: 1.23, 3.96), being overweight (AOR = 2.25, 95% CI: 1.20, 4.24), non-adherence to anti-hypertensive medication (AOR = 5.21, 95% CI: 2.76, 9.83), non-adherence to a low-salt diet and dietary approaches to stop hypertension (AOR = 2.74, 95% CI: 1.35, 5.53), taking three

**Funding:** The author(s) received no specific funding for this work.

**Competing interests:** The authors have declared that no competing interests exist.

**Abbreviations:** AOR, Adjusted Odds Ratio~; BMI, Body Mass Index; BP, Blood Pressure; BSc, Bachelor of Science; CI, Confidence Interval; CKD, Chronic Kidney Disease; COR, Crude Odds Ratio; CVD, Cardiovascular Disease; DASH, Dietary Approaches to Stop Hypertension; DBP, Diastolic Blood Pressure; HELM, Hypertension Evaluation Lifestyle Management; H-SCALE, Hypertension Self-Care Activity Level Effect; JNC-8, Eighth Joint National Committee; JNC-7, Seventh Joint National Committee; LMICs, Low and Middle-Income Countries; NCDs, Non-communicable Diseases; SBP, Systolic Blood Pressure; SPSS, Statistical Package for Social Science; WHO, World Health Organization.

or more antihypertensive medications (AOR = 3.10, 95% CI: 1.16, 8.25), and non-adherence to physical exercise (AOR = 2.84, 95% CI 1.49, 5.39) were factors associated with uncontrolled hypertension.

## Conclusions

Uncontrolled hypertension was very high in public hospitals in Addis Ababa, Ethiopia. Key factors for uncontrolled hypertension are non-adherence to antihypertensive medications, use of multiple medications, lack of physical exercise, and low adherence to low salt and dietary approaches to stop hypertension. To address these, enhancing patient education on medication adherence, promoting lifestyle changes, and leveraging digital health tools, like mobile apps, for real-time support and adherence tracking are recommended.

## Introduction

Cardiovascular disease (CVD) is the leading cause of death globally, responsible for 17.3 million deaths annually, with hypertension as a primary risk factor for nearly half of all CVD-related morbidity and mortality [1]. Hypertension is defined as having a systolic blood pressure (SBP) $\geq$ 140 mmHg and/or a diastolic blood pressure (DBP) $\geq$ 90 mmHg, measured on two separate days [2].

Hypertension is one of the avoidable risk factors for cardiovascular and cerebrovascular morbidity and mortality [3, 4]. Modifiable risk factors for hypertension include excessive sodium intake, low potassium intake, alcohol consumption, obesity, physical inactivity, and an unhealthy diet [5]. Hypertension prevalence has shifted from high-income countries to low- and middle-income countries (LMICs) [6].

Hypertension can be prevented and controlled through lifestyle modifications (weight reduction, the adoption of dietary approaches to stop hypertension (DASH) eating plans, sodium reduction, exercise, moderate alcohol intake, and smoking cessation), pharmacological treatments, and adherence to antihypertensive medications [7]. Failure to adhere to lifestyle modifications and pharmacologic therapy, which results in uncontrolled hypertension, is a serious public health problem in both developed and developing countries [8, 9].

In 2019, a global study found that only 23% of women and 18% of men on hypertension therapy effectively controlled their blood pressure [10]. In Malaysia, from 2006 to 2015, over 60% of hypertension cases were uncontrolled [11]. The pooled prevalence of uncontrolled hypertension in Ethiopia was 48% [12].

Key factors contributing to uncontrolled hypertension included non-adherence to antihypertensive medications, physical activity, alcohol abstinence, a low-salt and DASH diet, overweight/obesity, comorbidities, disease duration, sex, and age [9, 13, 14]. Furthermore, uncontrolled hypertension was associated with non-adherence to weight management and smoking abstinence, Khat chewing, marital status, income, residence, education level, awareness of complications, the number of antihypertensive medications, and family history of hypertension [9, 15–18].

A study in 12 health centers in Addis Ababa found a 69% prevalence of uncontrolled hypertension [19]. The magnitude varied from 54.9% at Tikur Anbessa Hospital to 73.8% at Zewditu Memorial Hospital [20, 21]. Most studies on uncontrolled hypertension in Ethiopia utilized the outdated 2003 guidelines. They classified patients' controlled and uncontrolled

hypertension using the joint national committee on prevention, detection, evaluation, and treatment of high blood pressure's seventh report (JNC-7) [22]. The updated 2014 evidence-based guidelines for the management of high blood pressure in adults reported by panel members appointed to the eighth joint national committee (JNC-8) recommended new cutoff points for target blood pressure goals on which classification is based [7].

Ethiopian studies on uncontrolled hypertension have shown significant disparities. Additionally, only a few factors were consistently identified across studies, with variations in associated factors. Non-adherence to alcohol abstinence, non-adherence to smoking abstinence, and Khat chewing were the factors that have shown inconsistency across different studies [8, 9, 13, 15, 23].

Uncontrolled hypertension varied significantly, from 11.4% to 73.8%. Most prior studies were conducted in a single hospital, and nearly half did not evaluate patients' knowledge of hypertension self-care and complications. Without assessing the knowledge status of patients, it will be difficult to fully understand patient-side factors for uncontrolled hypertension. Few studies have examined uncontrolled hypertension and its associated factors in Addis Ababa, Ethiopia's capital city, despite its distinct lifestyle as compared to the regions. This study aimed to assess the prevalence and factors contributing to uncontrolled hypertension among adult hypertensive patients in public hospitals in Addis Ababa, Ethiopia. This assessment used the updated JNC-8 guidelines, focusing on new blood pressure targets for antihypertensive treatment and incorporating patients' knowledge as a variable.

## Methods

### Study area and period

The study was conducted at selected public hospitals in Addis Ababa from April 12 to May 12, 2024. Addis Ababa comprises 11 sub-cities and 118 woredas and has an estimated total population of 3,854,863. This population includes an estimated 2,004,529 females and 1,850,334 males [24]. The study was conducted in three public hospitals in Addis Ababa, selected using simple random sampling methods. A total of 15,508 hypertensive patients were followed up for three months at these Addis Ababa public hospitals.

### Study design and population

We employed a hospital-based cross-sectional study design. The source population comprised all adult hypertensive patients receiving follow-up care at all public hospitals in Addis Ababa during the study period. The study population included all adult hypertensive patients in follow-up at selected public hospitals in Addis Ababa during the same period.

**Inclusion and exclusion criteria.** All hypertensive patients aged 18 years and older who had been receiving antihypertensive treatment for six months before the study period and were receiving treatment at outpatient follow-up during the study period were included. Pregnant mothers (due to potential pregnancy-related blood pressure changes), unconscious or critically ill patients who were unable to participate in an interview, and patients with incomplete medical records (specifically, those lacking one or two previous follow-up blood pressure measurements) were excluded from the study.

### Sample size determination and sampling procedures

The sample size for the study was calculated for both objectives (i.e., for specific objective 1 and specific objective 2), and the maximum of these calculated sample sizes was used. The sample size required for this study was calculated using a double proportion formula for

objective 2, based on the following assumptions: 80% study power, 37.9% proportion of non-adherence to physical exercise in exposed individuals, and 24.2% in unexposed individuals, a 95% confidence interval, and a 1:1 ratio of unexposed to exposed individuals [15]. The final sample size was determined after conducting a sensitivity analysis to assess whether a different sample size would yield different results. This calculation resulted in a required sample size of 425 participants. A simple random sampling lottery method was used to select the three study hospitals. Data were reviewed from the three hospitals over three months, encompassing 3552, 3690, and 3167 individuals who visited the chronic disease clinics of public hospitals in Addis Ababa. Then, the three-month flow was divided into three groups to obtain the average number of hypertensive patients at the one-month follow-up (1184, 1230, and 1056). We applied the proportional allocation method to determine the sample size, based on the number of hypertensive patients who visited the three public hospitals in March 2024 (the month preceding the survey). These hospitals comprised the source population (n = 3470), with patient counts of 1184, 1230, and 1056, respectively. The total sample size was then proportionally allocated to the selected hospitals according to the hypertensive patient flow. The final sample sizes for each hospital were 151, 145, and 129 (Fig 1).

During the data collection process, the study participants were chosen using systematic random sampling methods. Patient visits served as a sampling frame. For instance, we recruited patients based on their visit to a hypertension follow-up clinic using the sampling interval. The sampling interval (K) was determined by dividing the total number of hypertensive patients by the sample size (K = N/n = 3470/425 = 8), which means that the data were obtained from every eighth interval. The initial sample was selected randomly from their visit using a number between 1 and the sampling interval 8. The subsequent samples in the study were identified systematically through their visit to every eighth patient from the initial samples selected.

### Data collection tools and techniques

The data were collected using an interviewer-administered structured questionnaire through face-to-face interviews adapted from various studies [13, 15] using the hypertension self-care Activity level effect (H-SCALE) questionnaire [25] and the hypertension evaluation of lifestyle and management knowledge scale (HELM scale) [9, 26]. The H-SCALE is preferred because it offers a more comprehensive and detailed evaluation of a patient's status, as it includes medication adherence, physical activity, smoking, salt intake, alcohol, and weight management. By combining these different aspects, the H-SCALE paints a more complete picture of a patient's hypertension control as compared to relying solely on blood pressure readings. In addition, the HELM scale was used to supplement the knowledge of self-care, lifestyle management, and complications of the hypertension variable.

Modification of the scales was also performed to fit the local context, particularly in the DASH diet section, where some of the foods are not common in the Ethiopian context, and including them in the questionnaire is not relevant. For example, the question "eat processed meats such as ham, bacon, bologna, or sausage?" was excluded from the questionnaire, as these foods are not eaten in Ethiopia. The questionnaire was designed to gather information regarding socio-demographic variables, patient clinical variables, and behavioral practice-related variables. The internal consistency of the H-SCALE questionnaire was tested in studies performed in Eastern Ethiopia and at Jimma University and was reported to be acceptable, with good to excellent values. The specific values for each self-care domain were as follows: for medication adherence, 0.94; for a low-salt diet, 0.74; for physical activity adherence, 0.81; for weight management, 0.93; and for alcohol use, 0.92 [27].

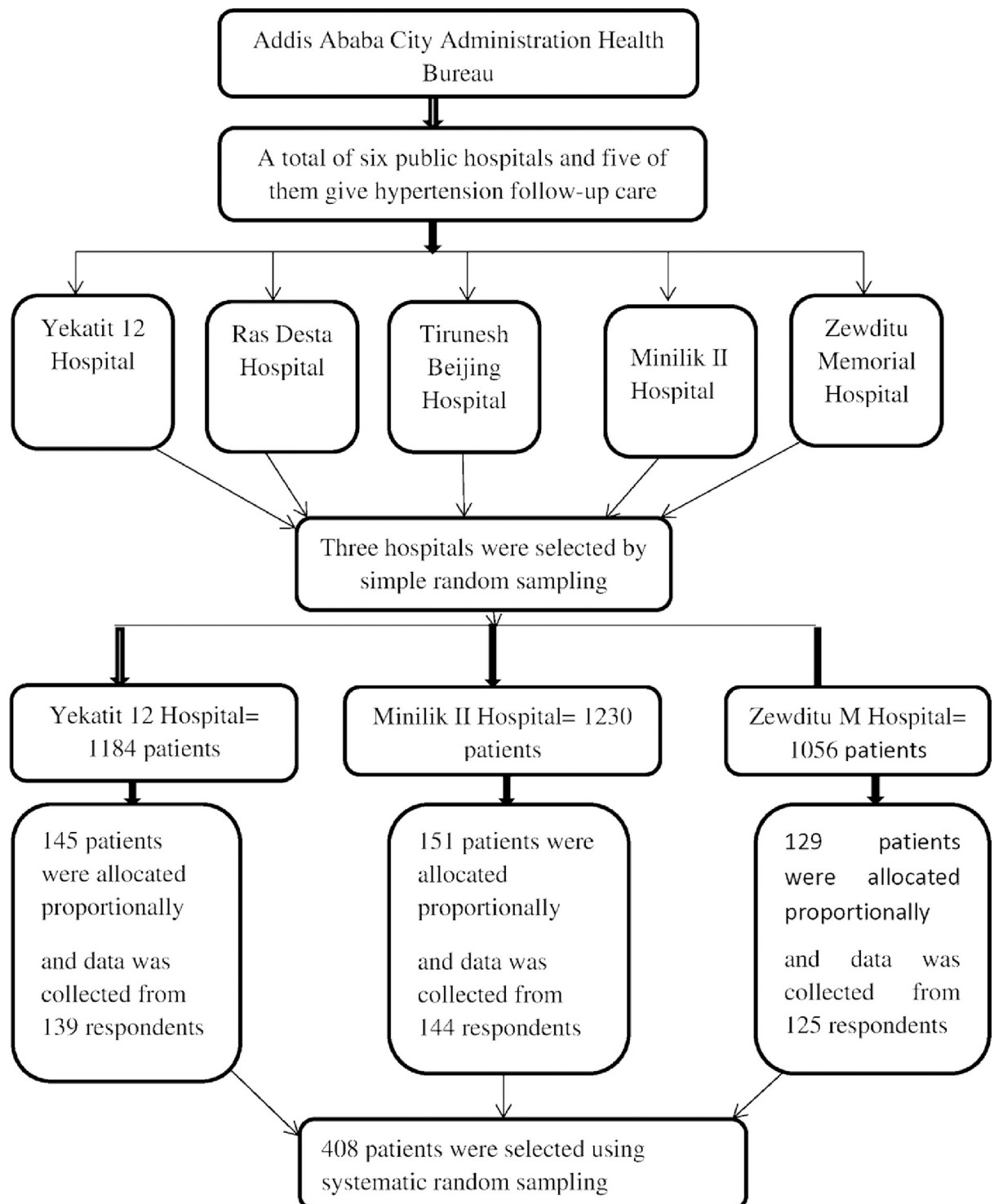

**Fig 1. Schematic diagram of the sampling procedure for the study of uncontrolled hypertension and associated factors among adult hypertensive patients on follow-up in public hospitals, Addis Ababa, Ethiopia, 2024.**

Variables such as khat chewing and knowledge of self-care practices and lifestyle management for hypertension patients were added after reviewing the literature [9]. The compiled questionnaire was translated into Amharic and back-translated to English to maintain internal consistency. Pretests were performed on 5% [21] of the eligible hypertensive patients at Ras desta Hospital before data collection to modify the data collection tool and evaluate the response rate. The sequence of the tool was modified based on the results of the pretest.

The data were collected by four BSc nurses for one month through close supervision and facilitation by two health officers and the principal investigator. In addition to face-to-face interviews, patient charts were reviewed to obtain data on previous BP records and comorbidity status. Finally, measurements of current blood pressure, height, and weight were taken on the day of data collection following standard procedures.

On the day of data collection, BP was measured using a standard sphygmomanometer BP cuff with an appropriate cuff size covering two-thirds of the upper arm with the patient in a sitting position. The sphygmomanometer was calibrated by biomedical engineers before the actual data collection to ensure the reliability and validity of the measurement. Additionally, the measurement was cross-checked with the other sphygmomanometers in the hospitals to check for discrepancies in measurement. The patient rested for five minutes and did not consume cigarettes or caffeine for thirty minutes. Additionally, excess clothing that could affect the BP cuff was removed. Patients were told to remain calm during BP measurements. The BP cuff was inflated enough to stop blood flow until no sound was heard through the stethoscope. Then, the cuff was deflated slowly to measure systolic and diastolic BP. Body weight was measured with light clothing using a digital weight scale, height was measured by removing shoes using a stadiometer, and BMI was calculated.

## Description of variables

**Dependent Variable**: uncontrolled hypertension.

Independent variables.

**Socio-demographic variables:** age, sex, marital status, educational level, monthly income in Ethiopian countries, residence status, and work status.

**Patient clinical variables** included the presence of comorbidities, current BMI, family history of hypertension, number of antihypertensive medications, and duration of hypertension diagnosis.

The behavioral practice-related variables included awareness of hypertension-related complications, adherence to antihypertensive medications, adherence to salt restriction, non-adherence to weight management, non-adherence to physical activity, non-adherence to alcohol abstinence, non-adherence to smoking abstinence, khat chewing, and current knowledge on self-care, lifestyle management, and complications of hypertension.

## Operational definition

**Uncontrolled hypertension** was defined as a patient's systolic blood pressure $\geq 150$ mmHg and/or diastolic blood pressure $\geq 90$ mmHg for patients aged $\geq 60$ years old or systolic blood pressure $\geq 140$ mmHg and/or diastolic blood pressure $\geq 90$ mmHg for all other patients aged $< 60$ years old who are on treatment, including patients with comorbidities [7]. Hypertensive patients were categorized as either uncontrolled or controlled based on whether they met blood pressure targets as defined by the JNC-8 guidelines.

**Medical comorbidity:** refers to a patient with hypertension who also has one or more other medical disorders. These conditions can be acute or chronic, and they can impact each other, sometimes in complex ways. Some of the comorbidities seen in a patient with hypertension

include the following: diabetes mellitus, chronic kidney disease, chronic obstructive pulmonary disease, coronary artery disease, dyslipidemia, sleep apnea, depression and anxiety, peripheral artery disease, atrial fibrillation, heart failure, previous stroke or transient ischemic attack, cognitive dysfunction, aortic valve stenosis [28–30].

The H-SCALE scoring method used to collect the data included the following variables [25]:

**Medication adherence:** Three items were used to assess the number of days in the preceding week that an individual took blood pressure medication, took it at the same time every day, and took the recommended dosage. Each item constituted 0–7 days as a response. Responses were summed (range 0–21) [25].

**Adherence to medication:** participants who reported that they followed these three recommendations on 7 out of 7 days (i.e., those who scored 21).

**Non-adherence to medications:** participants who scored 0–20.

Measuring adherence to medication and dietary recommendations is crucial in hypertension studies, as it directly impacts the effectiveness of interventions and overall outcomes. We employed self-reported measurements to measure medication adherence and low-salt diet adherence using questionnaires, interviews, and recall measures.

**Physical activity:** Physical activity was assessed by two items. 1. How many of the past seven days did you perform at least 30 minutes of total physical activity? 2. How many of the past seven days did you perform specific exercise activities (swimming, walking, or biking) other than what you did around the house or as part of the work? [25].

**Adherence to physical activity**: Participants who responded four days or more for each of two items or had a total score of $\geq 8$ were considered physically active [25].

**Smoking:** Smoking status was determined by one question: "how many cigarettes or cigars did you smoke in the last seven days, even if it was just one puff?" Respondents who reported zero days of smoking were classified as nonsmokers. Everyone else was classified as a smoker [25].

**Salt intake:** The H-SCALE has twelve items used to assess practices related to eating a healthy diet, avoiding salt while cooking and eating, and avoiding foods high in salt content in the preceding 7 days. The items were modified to nine, as items 3, 7, and 11 are country-specific. Six items were negatively phrased, and the responses for these items were reverse coded (responses were 0–7 days). Then, a mean score was calculated. A score of 6 or higher indicated that the participants followed a low-salt diet for 6 out of 7 days and were considered salt-restricted adherents [25].

**Alcohol:** Participants who reported not drinking any alcohol (tella, tej, katikalla, beer, and others) in the last 7 days or who reported not drinking at all were considered abstainers. All others were regarded as non-adherent [25].

**Weight management** - Ten items were used to assess weight management activities undertaken in the previous 30 days through dietary practices and physical activity. The response options ranged from strongly disagree (score = 1) to strongly agree (score = 5). Participants who agreed or strongly agreed with all ten items (score $\geq 40$) were classified as practicing good weight management habits (weight management adherent) [25].

**Overweight and obesity:** BMI was calculated and classified according to WHO guidelines. Participants were classified as underweight if their BMI was $< 18.5$ kg/m$^2$, overweight if their BMI was 25.0–29.99 kg/m2, and obese if their BMI was $\geq 30$ kg/m$^2$ [8].

**Awareness of hypertension-related complications:** This was assessed by one question with multiple response items [25].

**Aware:** participants who had at least two hypertension-related complications.

**Not aware:** those participants who responded either because they did not know about any hypertension-related complications or only one hypertension-related complication.

Knowledge of self-care, lifestyle management, and complications of hypertension was assessed by using the HELM scale, which contains 14 items to assess respondents' knowledge. The questions were modified to 10, as questions 7 and 8 were country-specific, and questions 12 and 13 did not meet the objective of the study. A correct response was recorded as 1, and an incorrect response was recorded as 0. The responses were summarized [26].

**Good knowledge:** participants who scored above or equal to the mean score of the HELM scale.

**Poor knowledge:** Participants who scored below the mean HELM score were considered to have poor knowledge.

**Data quality management.   Before data collection**: The English questionnaire was translated into the local language, Amharic, and then back into English to ensure consistency. Before data collection, the questionnaire was pretested on 5% of the total sample at Ras desta Hospital to determine the response rate, clarity, sequence, and consistency of the questionnaire. The sequence of the tool was adjusted based on the results of the pretest. An inter-rater reliability test was performed for nurses who were collecting the data using pilot testing, training, and standardization. The data collectors were given two days of training on the study's objective, relevance, and confidentiality of information, respondents' rights, informed consent, and interview techniques. In addition, a practical interview demonstration was held in a classroom.

**During data collection,** the data collection and interviewing mechanism was strictly supervised throughout the data collection period by the assigned supervisors and the principal investigator. The questionnaires were checked for completeness and consistency at the site of the data collection by the principal investigator.

**During data entry and analysis,** the collected data were coded and entered into Epidata software version 3.1. The quality of the data was controlled through skipping patterns that must enter and reduce transportation errors in Epidata. Finally, cleaning and analysis were performed using SPSS version 25.

## Data processing and analysis

The collected data were checked for completeness and then coded and entered into Epidata version 3.1 and exported to SPSS version 25 for further analysis. The outcome variables were classified into two groups, uncontrolled hypertension, and controlled hypertension, based on the average of three consecutive blood pressure measurements. The results of the analysis are presented in the form of text, tables, figures, and summary statistics. For categorical variables, frequencies, percentages, and figures were used. To identify the factors that were associated with uncontrolled hypertension, logistic regression was carried out. Variables with p-values $< 0.25$ in the bivariate analysis were candidates for multivariate analysis to control the effect of confounders. Multivariate analysis with a p-value $< 0.05$ was used to estimate associations between dependent and independent variables.

## Ethics approval and consent to participate

Ethical approval was obtained from the Yekatit 12 Hospital Medical College department of public health IRB [Ref.no. Rpo/130/22], Addis Ababa, Ethiopia, on 13 May 2022. Written informed consent was obtained from each study participant to ensure willingness. Information about the benefits and harms of the study, the usefulness of their participation, the confidentiality of the information, and the right not to participate were given to the participants. The data collectors were given two days of training on the study's objective, relevance, and confidentiality of information, respondents' rights, informed consent, and interview techniques to

address potential ethical vulnerabilities of patients not to feel obliged to participate in the study. Data for patients' clinical variables like blood pressure measurements, comorbidities, and number of anti-hypertensive medications was obtained by accessing personal health records during their visit to the hypertension follow-up clinic after obtaining written consent. All the procedures utilized to access personal health data were regulated by the personal data Protection act law and regulation to avoid any potential breach of data privacy.

## Results

### Socio-demographic characteristics

A total of 408 patients participated in the study, which is a response rate of 96%. More than half (52.7%) of the respondents were females. The mean (± SD) age of the respondents was 56.26 (± 13.46) years. More than half (56.1%) of the respondents were younger than 60 years. The majority of the respondents (65.4%) were married. One hundred twelve (27.5%) and 110 (27%) had no formal education and a college education or above, respectively (Table 1).

**Clinical characteristics of the respondents.** Nearly half of the respondents (41.9%) had a normal BMI, followed by overweight (39%). In addition, the mean (± SD) BMI was 26.08 (± 4.28) kg/m$^2$. The mean (± SD) weight of the respondents was 69.4 (± 11.39) kilograms, and the mean (± SD) height was 1.63 (± 0.08) meters (Table 2).

**Knowledge of respondents.** More than half of the respondents (51%) reported having at least two hypertension-related complications, while the rest did not. Concerning knowledge of self-care and lifestyle management for hypertension patients, 62.3% of the respondents had

**Table 1. Socio-demographic characteristics of adult hypertensive patients on follow-up in public hospitals of Addis Ababa, Ethiopia, 2024 (n = 408).**

| Variable | Category | Frequency | Percent (%) |
|---|---|---|---|
| Sex | Male | 193 | 47.3 |
| | Female | 215 | 52.7 |
| Age category | <60 years | 229 | 56.1 |
| | ≥60 years | 179 | 43.9 |
| Marital status | Married | 267 | 65.4 |
| | Single | 31 | 7.6 |
| | Divorced | 33 | 8.1 |
| | Widowed | 77 | 18.9 |
| Educational status | No formal education | 112 | 27.5 |
| | Primary | 102 | 25 |
| | Secondary | 84 | 20.6 |
| | College and above | 110 | 27 |
| Occupational status | Housewife | 101 | 24.8 |
| | Government employee | 75 | 18.4 |
| | Unemployed | 30 | 7.4 |
| | Retired | 93 | 22.8 |
| | Private business | 106 | 26 |
| | Other * | 3 | 0.7 |
| Income category | ≤ 500 ETB | 15 | 3.7 |
| | 501–1000 ETB | 44 | 10.8 |
| | >1000 ETB | 349 | 85.5 |

*Student, daily laborer

**Table 2. Clinical characteristics of adult hypertensive patients on follow-up in public hospitals of Addis Ababa, Ethiopia, 2024 (n = 408).**

| Variable | Category | Frequency | Percent (%) |
|---|---|---|---|
| BMI | Underweight | 7 | 1.7 |
| | Normal weight | 171 | 41.9 |
| | Overweight | 159 | 39 |
| | Obese | 71 | 17.4 |
| Number of antihypertensive medications | ≤ 2 medications | 335 | 82.1 |
| | Three or more | 73 | 17.9 |
| Duration of hypertension diagnosis | < 5 years | 161 | 39.5 |
| | 5–10 years | 121 | 29.7 |
| | ≥ 10 years | 126 | 30.9 |
| Family history of hypertension | Yes | 196 | 48 |
| | No | 212 | 52 |
| Comorbidity status | Yes | 211 | 51.7 |
| | No | 197 | 48.3 |

good knowledge regarding lifestyle management for hypertension patients. The mean (± SD) score of knowledge was 5.07 (± 2), with a maximum score of 10 and a minimum score of 0.

## Behavioral practices of the respondents

Half of the respondents (51.2%) did not adhere to antihypertensive medications (e.g., BP pills, the right time, and the recommended dosage). The majority of the respondents (60.5%) did not adhere to physical activity (regular activity or specific activity). More than three-fourths (79.4%) of the respondents did not adhere to the low-salt and DASH diets (Table 3).

## Prevalence of uncontrolled hypertension

The prevalence of uncontrolled hypertension among adult hypertensive patients in public hospitals in Addis Ababa was 66.2% (95% CI = 61.61%, 70.79%). The mean (± SD) SBP of the average of three consecutive blood pressure measurements was 146.16 (± 15.3) mmHg, with a maximum SBP of 188.67 mmHg and a minimum SBP of 106.67 mmHg. The mean (± SD)

**Table 3. Behavioral practice-related variables of adult hypertensive patients on follow-up in public hospitals of Addis Ababa, Ethiopia, 2024 (n = 408).**

| Variable | Category | Frequency | Percent (%) |
|---|---|---|---|
| Khat chewing | Yes | 21 | 5.1 |
| | No | 387 | 94.9 |
| Smoking status | Yes | 15 | 3.7 |
| | No | 393 | 96.3 |
| Alcohol drinking | Yes | 21 | 5.1 |
| | No | 387 | 94.9 |
| Medication Adherence | Adhere | 199 | 48.8 |
| | Not adhere | 209 | 51.2 |
| Physical Exercise Adherence | Adhere | 247 | 60.5 |
| | Not adhere | 161 | 39.5 |
| Low salt diet | Adhere | 84 | 20.6 |
| | Not adhere | 324 | 79.4 |
| Adherence to weight management practices | Adhere | 143 | 35 |
| | Not adhere | 265 | 65 |

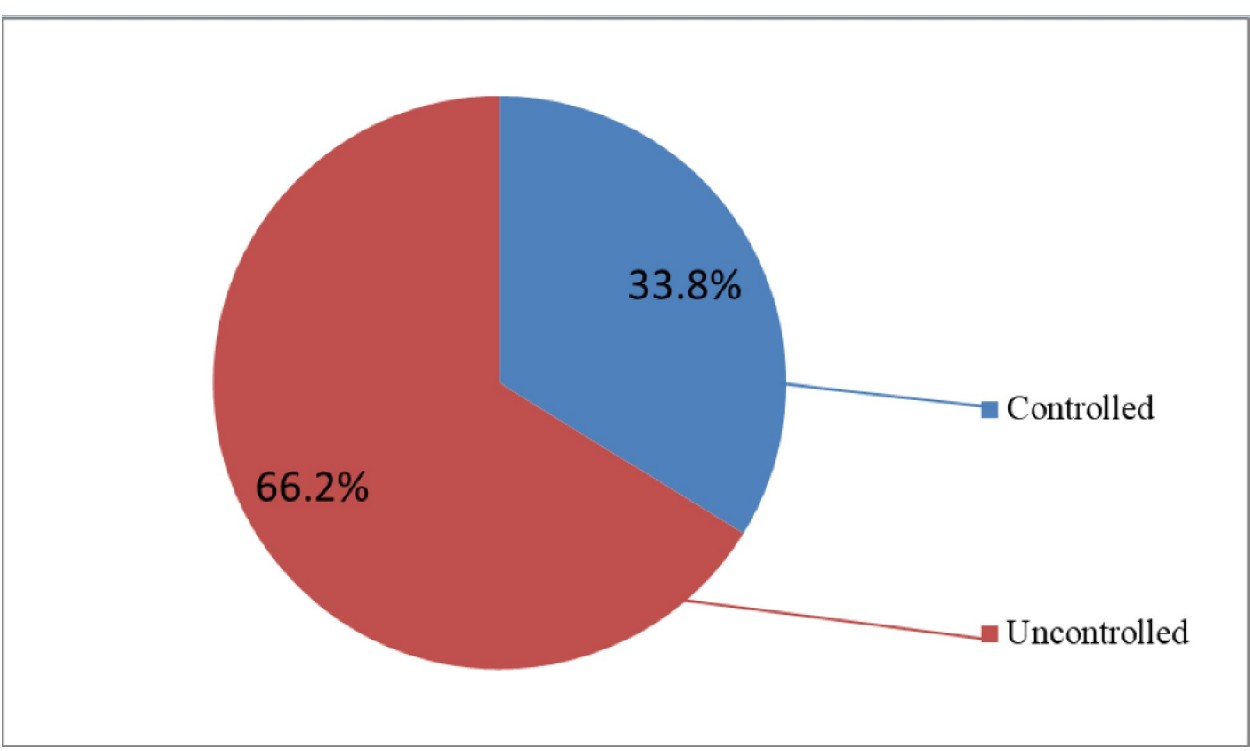

**Fig 2. Magnitude of uncontrolled hypertension among adult hypertensive patients on follow-up in public hospitals of Addis Ababa, Ethiopia, 2024 (n = 408).**

DBP of the average of three consecutive blood pressure measurements was 84.27 (±8.18) mmHg, with a maximum DBP value of 100 mmHg and a minimum of 60.67 mmHg (Fig 2).

### Factors associated with uncontrolled hypertension

After adjusted analysis, the presence of comorbidities, age ≥ 60 years, overweight status, no adherence to antihypertensive medications, no adherence to low-salt and DASH diets, taking three or more antihypertensive medications, and no adherence to physical exercise were associated with uncontrolled hypertension.

Hypertensive patients aged 60 years and above were nearly three times (AOR = 2.88, 95% CI: 1.37, 6.04) more likely to develop uncontrolled hypertension than those aged less than 60 years. Hypertensive patients with one or more comorbid conditions were approximately two times more likely to develop uncontrolled hypertension than those who were not diagnosed with comorbid conditions (AOR = 2.21, 95% CI: 1.23, 3.96). In this study, overweight hypertensive patients were 2.25 times more likely to develop uncontrolled hypertension than normal-weight patients (AOR = 2.25, 95% CI: 1.20, 4.24).

Hypertensive patients who did not adhere to their antihypertensive medications were 5.21 times more likely to have uncontrolled hypertension as compared to their counterparts (AOR = 5.21, 95% CI: 2.76, 9.83). Those who did not adhere to a low-salt diet or DASH diet were approximately three times more likely to develop uncontrolled hypertension than those who did adhere to a low-salt or DASH diet (AOR = 2.74, 95% CI: 1.35, 5.53). Hypertensive patients who were taking three or more antihypertensive medications were 3.1 times more likely to develop uncontrolled hypertension than those who were taking one or two antihypertensive medications (AOR = 3.10, 95% CI: 1.16, 8.25). Those who did not adhere to physical

exercise were 2.84 times more likely to have uncontrolled hypertension than those who did (AOR = 2.84, 95% CI: 1.49, 5.39) (Table 4).

## Discussion

The study revealed that nearly two-thirds of hypertensive patients who were taking anti-hypertensive treatment had uncontrolled hypertension. Age $\geq$ 60 years, the presence of comorbidities, being overweight, not adhering to antihypertensive medications, not adhering to low-salt and DASH diets, taking three or more antihypertensive medications, and not adhering to physical exercise were factors associated with uncontrolled hypertension.

Around two-thirds of hypertensive patients who were taking antihypertensive treatment had uncontrolled hypertension. This prevalence is consistent with findings from a study conducted in six Latin American countries (Argentina, Brazil, Chile, Colombia, Peru, and Uruguay) (62.4%) [31], the Islamic Republic of Iran (68.5%) [32], India (68.3%) [33], Republic of Congo (66%) [34], Addis Ababa Health Centers (69%) [19], and Zewditu Memorial Hospital (69.9%) [35].

However, this prevalence is lower than that reported in the Peru demographic health survey (94.7%) [36], Afghanistan (77.3%) [37], Morocco (73%) [17], and Zewditu Memorial Hospital (73.8%) [20]. The disparity in the findings of the Zewditu Memorial Hospital study may be due to the smaller sample size used in the study than in the present study. Furthermore, the study was limited to a single facility, which could lead to overestimation. Additionally, the average blood pressure level was calculated after only two consecutive measurements were taken in a single visit, four minutes apart from those in the previous study.

Uncontrolled hypertension in this study is higher than in studies done in Albania (48.4%) [38], Russia (47.8%) and Norway (38.2%) [39], in Thailand (24.6%) [18], other study in Central areas of Thailand (54.4%) [40], Iran (61.1%) [41], Bangladesh, Pakistan, Sri Lanka (58%) [16], Shanghai, China (56.6%) [42], Nepal (48%) [43], South Africa (56.83%) [44], Ghana (57.7%) [45], Botswana (55%) [46], Cameroon (57.2%) [47], Tanzania (37.2%) [48], Eastern Sudan (54.7%) [49], Gonder University Hospital (49.6%) [50], Debre Tabor (57.1%) [51], Jimma University (52.7%) [9], Ayder Comprehensive Specialized Hospital (52.5%) [13], Mekelle Hospitals (48.6%) [52] and Nekemte (36.4%) [15]. The relatively higher magnitude in this study could be because it was a population-based study in Russia and Norway and a nationwide survey in Thailand. Differences in socioeconomic status and degree of urbanization may also play a role in this discrepancy. Furthermore, the difference in lifestyle between Addis Ababa and the rest of Ethiopia may contribute to this increase. The majority of the studies were also conducted using the JNC-7 guidelines for hypertension classification and control. The level of adherence to antihypertensive medications was greater in a study performed in Ayder than in the present study, which could have contributed to this finding.

In conclusion, the prevalence of uncontrolled hypertension in LMICs ranges from 55% to 77.3%, which is almost in line with the prevalence of uncontrolled hypertension from this study [17, 34, 37, 44–47].

Among the socio-demographic and economic variables, age was significantly associated with uncontrolled hypertension. In this study, hypertensive patients aged sixty years and older were 2.88 times more likely to develop uncontrolled hypertension than those aged younger than sixty years (AOR = 2.88). This finding is similar to that of an Iranian study [32], a study conducted in three Mekelle hospitals [52], and a study at Jimma University [9]. This could be because as people age increased, their blood vessels lose elasticity, resulting in peripheral vascular resistance and uncontrolled hypertension [9].

In this study, hypertensive patients with comorbidities were 2.21 times more likely to have uncontrolled hypertension than those who were not diagnosed with comorbidities. This finding is

**Table 4. Bivariate and multivariate logistic regression to identify factors associated with uncontrolled hypertension among adult hypertensive patients on follow-up in public hospitals of Addis Ababa, Ethiopia, 2024 (n = 408).**

| Variable | Category | Uncontrolled hypertension | | COR (95% CI) | p-value | AOR (95% CI) | p-value |
|---|---|---|---|---|---|---|---|
| | | Yes (%) | No (%) | | | | |
| Sex | Male | 136 (33.3) | 57 (14) | 1.44 (0.95, 2.18) | 0.083 | 1.10 (0.57, 2.12) | 0.788 |
| | Female | 134 (32.8) | 81 (19.9) | 1 | | 1 | |
| Marital status | Married | 171 (41.9) | 96 (23.5) | 1 | | 1 | |
| | Single | 16 (3.9) | 15 (3.7) | 0.60 (0.28, 1.26) | 0.179 | 1.14 (0.41, 3.15) | 0.806 |
| | Divorced | 21 (5.1) | 12 (2.9) | 0.98 (0.46, 2.08) | 0.963 | 0.67 (0.24, 1.86) | 0.446 |
| | Widowed | 62 (15.2) | 15 (3.7) | 2.32 (1.25, 4.3) | 0.007 | 1.78 (0.74, 4.30) | 0.198 |
| Comorbidity | Yes | 166 (40.7) | 45 (11) | 3.30 (2.14, 5.08) | 0.000 | 2.21 (1.23, 3.96) | 0.008 |
| | No | 104 (25.5) | 93 (22.8) | 1 | | 1 | |
| Khat chewing | Yes | 18 (4.4) | 3 (0.7) | 3.21 (0.93, 11.11) | 0.065 | 2.19 (0.50, 9.60) | 0.299 |
| | No | 252 (61.8) | 135 (33.1) | 1 | | 1 | |
| Age category | <60 years | 127 (31.1) | 102 (25) | 1 | | 1 | |
| | ≥60 years | 143 (35) | 36 (8.8) | 3.19 (2.04, 5.00) | 0.000 | 2.88 (1.37, 6.04) | 0.005 |
| Educational status | No formal education | 88 (21.6) | 24 (5.9) | 3.41 (1.90, 6.13) | 0.000 | 0.97 (0.36, 2.61) | 0.944 |
| | Primary | 65 (15.9) | 37 (9.1) | 1.63 (0.94, 2.83) | 0.080 | 0.66 (0.28, 1.52) | 0.327 |
| | Secondary | 60 (14.7) | 24 (5.9) | 2.33 (1.27, 4.25) | 0.006 | 1.63 (0.72, 3.71) | 0.244 |
| | College and above | 57 (14) | 53 (13) | 1 | | 1 | |
| Occupational category | Housewife | 65 (15.9) | 36 (8.8) | 1 | | 1 | |
| | Governmental employee | 43 (10.5) | 32 (7.8) | 0.74 (0.40, 1.37) | 0.345 | 1.34 (0.50, 3.61) | 0.567 |
| | Unemployed | 15 (3.7) | 15 (3.7) | 0.55 (0.24, 1.26) | 0.160 | 0.74 (0.21, 2.63) | 0.641 |
| | Retired | 76 (18.6) | 17 (4.2) | 2.48 (1.27, 4.82) | 0.008 | 2.10 (0.72, 6.15) | 0.177 |
| | Private business | 70 (17.2) | 36 (8.8) | 1.08 (0.61, 1.91) | 0.800 | 1.73 (0.68, 4.41) | 0.254 |
| | Other | 1 (0.2) | 2 (0.5) | 0.28 (0.02, 3.16) | 0.301 | 1.26 (0.08, 19.49) | 0.868 |
| BMI category | Normal weight | 86 (21.1) | 85 (20.8) | 1 | | 1 | |
| | Overweight | 127 (31.1) | 32 (7.8) | 3.92 (2.40, 6.40) | 0.000 | 2.25 (1.20, 4.24) | 0.012 |
| | Obese | 54 (13.2) | 17 (4.2) | 3.14 (1.69, 5.85) | 0.000 | 1.95 (0.87, 4.35) | 0.103 |
| | Underweight | 3 (0.7) | 4 (1) | 0.74 (0.16, 3.41) | 0.701 | 2.56 (0.36, 18.32) | 0.350 |
| Duration of HTN diagnosis | <5 years | 95 (23.3) | 66 (16.2) | 1 | | 1 | |
| | 5–10 years | 72 (17.6) | 49 (12) | 1.02 (0.63, 1.65) | 0.933 | 0.72 (0.37, 1.39) | 0.320 |
| | ≥10 years | 103 (25.2) | 23 (5.6) | 3.11 (1.79, 5.40) | 0.000 | 0.82 (0.37, 1.82) | 0.622 |
| Medication adherence | Yes | 95 (23.3) | 104 (25.5) | 1 | | 1 | |
| | No | 175 (42.9) | 34 (8.3) | 5.64 (3.55, 8.93) | 0.000 | 5.21 (2.76, 9.83) | 0.000 |
| Low-salt diet adherence | Yes | 42 (10.3) | 42 (10.3) | 1 | | 1 | |
| | No | 228 (55.9) | 96 (23.5) | 2.38 (1.46, 3.88) | 0.001 | 2.74 (1.35, 5.53) | 0.005 |
| Weight mgt | Yes | 83 (20.3) | 60 (14.7) | 1 | | 1 | |
| | No | 187 (45.8) | 78 (19.1) | 1.73 (1.13, 2.65) | 0.011 | 1.85 (0.99, 3.44) | 0.052 |
| Number of anti-hypertensive | ≤ two | 205 (50.2) | 130 (31.9) | 1 | | 1 | |
| | ≥ three | 65 (15.9) | 8 (2) | 5.15 (2.39, 11.09) | 0.000 | 3.10 (1.16, 8.25) | 0.024 |
| Physical exercise | Adhere | 71 (17.4) | 90 (22.1) | 1 | | 1 | |
| | Not adhere | 199 (48.8) | 48 (11.8) | 5.26 (3.38, 8.18) | 0.000 | 2.84 (1.49, 5.39) | 0.001 |
| Monthly income category | ≤500 | 8 (2) | 7 (1.7) | 0.61 (0.22, 1.73) | 0.357 | 0.51 (0.10, 2.52) | 0.409 |
| | 501–1000 | 35 (8.6) | 9 (2.2) | 2.09 (0.97, 4.49) | 0.059 | 1.41 (0.49, 4.08) | 0.522 |
| | >1000 | 227 (55.6) | 122 (29.9) | 1 | | 1 | |

consistent with the findings of a study conducted at Ayder Comprehensive Specialized Hospital [13]. A possible explanation could be that many chronic diseases can cause secondary hypertension, making it difficult to control hypertension while also treating other illnesses [13].

The odds of having uncontrolled hypertension were 2.25 times greater in overweight hypertensive patients than in normal-weight hypertensive patients in this study. This finding is consistent with those of a Moroccan study [17], Ayder Comprehensive Specialized Hospital [13], and Jimma University study [9]. This could be explained by the fact that being overweight increases afterload and peripheral vascular resistance, which leads to increased triglyceride and cholesterol levels, decreased high-density lipoprotein (HDL) levels, and uncontrolled hypertension [9].

Medication adherence is one of the crucial things in controlling hypertension. In this study, 51.2% of the hypertensive patients were not adhering to their anti-hypertensive medications. This finding is higher than studies conducted in Afghanistan (42.1%) [37], Ayder Comprehensive Specialized Hospital (25.9%) [13], Jimma (40.6%) [9], and Bale Zone Hospitals (39%) [8], and in Mekelle public hospitals (26.1%) [52]. The most possible reason could be taking multiple medications, leading to pill burden, increased risk of side effects, and cost concerns [53]. In addition, it could be due to low motivation due to side effects, belief in the effectiveness of treatment, insufficient patient-provider communication, or higher co-payments [54, 55]. Furthermore, it could be due to having comorbidities, sociocultural and financial reasons, and the complexity of the regimen [56]. The magnitude of medication non-adherence in this study is almost in line with the findings from studies done in global cities like Riyadh (57.8%) [57], Asmara (72.8%) [58], and Kandahar City, Afghanistan (47.9%) [59].

This study revealed that the odds of having uncontrolled hypertension were 5.21 times greater for those who did not adhere to antihypertensive medication than for those who did. This finding is consistent with studies performed in Bangladesh, Pakistan, Sri Lanka [16], Afghanistan [37], Ghana [45], Eastern Sudan [49], Ayder Comprehensive Specialized Hospital [13, 23], three Mekelle Hospitals [52] and Nekemte [15]. This consistency could be attributed to the fact that good medication adherence is critical for controlling high blood pressure through vasodilation, sodium, and fluid reduction through increased urination and antagonizing sympathetic activation of the heart [52].

Similarly, the study revealed that 79.4% of hypertensive patients were not adhering to a low-salt and DASH diet. This finding is higher than studies conducted in Ayder Comprehensive Specialized Hospital (36.9%) [13]. This finding is lower than that of studies conducted in Mekelle public hospitals (73.9%) [52].

In this study, hypertensive patients who did not follow a low-salt and DASH diet were 2.74 times more likely to have uncontrolled hypertension than those who follow. This finding is consistent with studies performed at Gonder University Hospital [60], Ayder Comprehensive Specialized Hospital [23], and three Mekelle Hospitals [52]. This similarity could be explained by the fact that salt reduces the natural sodium balance in the body, causing fluid retention and, as a result, increasing the pressure exerted by the blood vessel walls, leading to uncontrolled hypertension [50].

This study also revealed that hypertensive patients who were taking three or more antihypertensive medications were 3.1 times more likely to have uncontrolled hypertension than those who were taking one or two antihypertensive medications. This finding is in line with studies performed nationwide studies in Thailand [18], Ghana [45], and Ayder Comprehensive Specialized Hospital [23]. This consistency could be due to the possibility that those on three or more antihypertensive medications are present because they are unable to control their blood pressure with one or two antihypertensive medications and are still unable to control it with three or more drugs [23].

Finally, the study revealed that hypertensive patients who did not adhere to regular physical exercise were 2.84 times more likely to develop uncontrolled hypertension than those who did regular physical exercise. This finding is consistent with studies performed at Ayder

Comprehensive Specialized Hospital [13], Three Mekelle Hospitals [52], and Nekemt [15]. These findings may be consistent because regular physical activity helps to strengthen the heart, allowing it to pump more blood with less effort. If the heart pumps blood with less effort, then the force on the arteries decreases and blood pressure is controlled. Another mechanism by which physical exercise decreases blood pressure among hypertensive patients is through a reduction in systemic vascular resistance, autonomic nervous system function, plasma norepinephrine, insulin sensitivity, and renin activity. Furthermore, exercise lowers blood pressure by decreasing body weight and increasing renal function [15].

## Limitations of the study

This study has several limitations. The study might have a social desirability bias on self-reported sensitive issues like cigarette smoking, alcohol drinking, and Khat chewing, leading to classifying them as abstinence. Since most of the H-SCALE questions are self-reported and participants may tend to respond in a manner that will be viewed favorably by others, social desirability bias might be present. For example, for alcohol drinking, the participants might respond as if they were alcohol abstainers even if they drank alcohol. Recall bias could also be a limitation since the tool used to measure the self-care variables was H-SCALE and the technique was self-reported interview. Recall bias could also be present as the tool was self-reported and it was dependent on their memory; some recall bias might happen. As the study used a cross-sectional study design, it is difficult to relate the temporal relationship. It also has a short duration of study.

## Conclusion

The study revealed that a high prevalence of uncontrolled hypertension among hypertensive patients in Addis Ababa public hospitals, with non-adherence to antihypertensive medications as a key predictor. Other contributing factors include use of multiple medications, lack of physical exercise, and low adherence to low-salt and DASH diets. Prioritizing targeted interventions to improve antihypertensive medication adherence, including structured follow-ups, enhanced patient education on adherence benefits, and the use of digital health tools like mobile apps for tracking and support.

**Multidisciplinary Care Models:** We should advocate for a team-based care approach involving physicians, nurses, pharmacists, and community health workers, given the multifactorial nature of hypertension.

**Patient-Centered Approaches**: Empowering patients to actively manage their hypertension through involvement in treatment decisions and personalized care.

**Policy Advocacy**: Broader community and policy initiatives should include national campaigns to raise awareness of medication adherence and financial incentives for antihypertensive medications to improve access for low-income populations.

## Acknowledgments

We would like to thank the data collectors and study participants for their cooperation in the process of data collection and provision of their information.

## Author Contributions

**Conceptualization:** Asmamaw Deguale Worku, Asinake Wudu Gessese.

**Data curation:** Asmamaw Deguale Worku, Asinake Wudu Gessese.

**Formal analysis:** Asmamaw Deguale Worku.

**Methodology:** Asmamaw Deguale Worku, Asinake Wudu Gessese.

**Software:** Asmamaw Deguale Worku.

**Supervision:** Asmamaw Deguale Worku.

**Validation:** Asinake Wudu Gessese.

**Visualization:** Asinake Wudu Gessese.

**Writing – original draft:** Asmamaw Deguale Worku.

**Writing – review & editing:** Asmamaw Deguale Worku.

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
