## [Decision Letter · Decision Letter 0]

27 Sep 2024

PONE-D-24-37212Prevalence and associated factors of uncontrolled hypertension among adult hypertensive patients at public hospitals in Addis Ababa, Ethiopia : a cross- sectional study designPLOS ONE

Dear Dr. Worku,

Thank you for submitting your manuscript to PLOS ONE. After careful consideration, we feel that it has merit but does not fully meet PLOS ONE’s publication criteria as it currently stands. Therefore, we invite you to submit a revised version of the manuscript that addresses the points raised during the review process.

Two reports have been obtained. Please find these below. In your revision, please consider my suggestions as well as the suggestions provided by reviewers, particularly Reviewer 2, for further consideration.==============================

We look forward to receiving your revised manuscript.

Kind regards,

Muhammad Haroon Stanikzai

Academic Editor

PLOS ONE

Journal Requirements:

2. In the online submission form, you indicated that “The datasets used and/or analyzed during the current study are available from the corresponding author upon reasonable request.”

All PLOS journals now require all data underlying the findings described in their manuscript to be freely available to other researchers, either 1. In a public repository, 2. Within the manuscript itself, or 3. Uploaded as supplementary information. This policy applies to all data except where public deposition would breach compliance with the protocol approved by your research ethics board. If your data cannot be made publicly available for ethical or legal reasons (e.g., public availability would compromise patient privacy), please explain your reasons on resubmission and your exemption request will be escalated for approval.

4. Please include your tables as part of your main manuscript and remove the individual files. Please note that supplementary tables (should remain/ be uploaded) as separate "supporting information" files"

5. We note that there is identifying data in the Supporting Information file <Table 1.docx and Table 4.docx>. Due to the inclusion of these potentially identifying data, we have removed this file from your file inventory. Prior to sharing human research participant data, authors should consult with an ethics committee to ensure data are shared in accordance with participant consent and all applicable local laws. Data sharing should never compromise participant privacy. It is therefore not appropriate to publicly share personally identifiable data on human research participants. The following are examples of data that should not be shared: -Name, initials, physical address -Ages more specific than whole numbers -Internet protocol (IP) address -Specific dates (birth dates, death dates, examination dates, etc.) -Contact information such as phone number or email address -Location data -ID numbers that seem specific (long numbers, include initials, titled “Hospital ID”) rather than random (small numbers in numerical order) Data that are not directly identifying may also be inappropriate to share, as in combination they can become identifying. For example, data collected from a small group of participants, vulnerable populations, or private groups should not be shared if they involve indirect identifiers (such as sex, ethnicity, location, etc.) that may risk the identification of study participants. Additional guidance on preparing raw data for publication can be found in our Data Policy (https://journals.plos.org/plosone/s/data-availability#loc-human-research-participant-data-and-other-sensitive-data) and in the following article: http://www.bmj.com/content/340/bmj.c181.long. Please remove or anonymize all personal information (<specific identifying information in file to be removed>), ensure that the data shared are in accordance with participant consent, and re-upload a fully anonymized data set. Please note that spreadsheet columns with personal information must be removed and not hidden as all hidden columns will appear in the published file.

Additional Editor Comments:

- Although at times the writing is strong and clear, throughout most of the document the writing is circuitous, unclear and contains gramma1cal errors that obscure meaning. Recommend the entire paper be edited for conciseness, clarity and grammar before re-submission.

-In Abstract add the full form of DASH when its first used.

- Introduction: The Introduction is a major issue. As the first part of the article that a potential reader reads, it should be attractive and easy to read. The authors can make it concise. A lot of information such as hypertension definition, and types of anti-hypertensive medications can be removed. Definitions of uncontrolled hypertension should be moved to methods. The details of previous studies should be presented in very brief. Are there any current interventions in the study area for uncontrolled hypertension (Authors can add a brief paragraph). I propose the authors use, consult, and add the following reference in the manuscript.

- https://www.dovepress.com/high-prevalence-of-uncontrolled-hypertension-among-afghan-hypertensive-peer-reviewed-fulltext-article-IBPC

- Methods:

- Lines 177-178: Difficult to understand. Please rephrase

- Line 187: Please add a space between citation and text. Please follow this through all manuscript.

- Line 209: Please merge both citations. Please follow this through all manuscript.

- Line 253: Uncontrolled hypertension definition: Please revise as currently patient with uncontrolled hypertension not uncontrolled hypertension.

- Lines 257-260: Please add references to medication adherence and other independent variables where applicable.

- Lines 389-391: Difficult to understand. Please rephrase

-Discussion:

- The authors can summarize the prevalence of uncontrolled hypertension when compared with other LMICs. At current form it is difficult to understand. They can revise us the prevalence of uncontrolled hypertension in LMICs ranges from .......

- Recommend that authors re-write the discussion by condensing the informa0on into well-formed paragraphs that link together to form a story of what they recommend from each of the key observations they have made.

- Conclusion: Conclusions would benefit from more concise language and clear, specific study-based recommenda6ons

- Please include Tables in the main manuscript file.

Reviewers' comments:

Reviewer's Responses to Questions

**Comments to the Author**

1. Is the manuscript technically sound, and do the data support the conclusions?

Reviewer #1: Partly

Reviewer #2: Partly

2. Has the statistical analysis been performed appropriately and rigorously? 

Reviewer #1: Yes

Reviewer #2: Yes

3. Have the authors made all data underlying the findings in their manuscript fully available?

Reviewer #1: Yes

Reviewer #2: Yes

4. Is the manuscript presented in an intelligible fashion and written in standard English?

Reviewer #1: Yes

Reviewer #2: Yes

5. Review Comments to the Author

Reviewer #1: First and foremost, thank you for the opportunity to review your manuscript.

This feedback is intended to assist the authors in refining their work to meet high academic standards and ensure that their findings contribute effectively to the existing body of knowledge on uncontrolled hypertension.

I look forward to the authors' revisions which incorporate these suggestions to strengthen the methodological rigor and ethical transparency of the study.

These suggestions are intended to improve the clarity, ethical integrity, and methodological transparency of the manuscript, thus strengthening its overall scientific contribution.

By addressing the following aspects, the authors can ensure that the study is scientifically rigorous and ethically compliant, providing transparency in both variable measurement and data protection.

More details please go to the attached.

Reviewer #2: Thank you, editorial team, for inviting me to review this manuscript. I am grateful for the opportunity to review such an important topic, titled "Prevalence and associated factors of uncontrolled hypertension among adult hypertensive patients at public hospitals in Addis Ababa, Ethiopia: a cross-sectional study design."

Below are my comments and suggestions for the authors to help improve the manuscript:

1. Title: The title is informative but could be more concise. Consider simplifying it to: "Uncontrolled Hypertension in Adult Patients in Addis Ababa Public Hospitals: Prevalence and Associated Factors."

2. Abstract (Lines 15–42):

The abstract provides a good summary, but it can be improved by clarifying certain points. For example, instead of saying, "evidence regarding uncontrolled hypertension is very limited," be more specific about the gap in the literature.

3. I have read many studies conducted on a similar topic in Ethiopia, which indicated prevalence rates of 48.6% in North Ethiopia, 48% in Eastern Ethiopia, 52.7% in Western Ethiopia, and 52.5% in the Tigray region. What distinguishes your research from these studies?

4. Introduction (Lines 45–154):

Comment: The introduction effectively frames the issue of hypertension, but it could benefit from some tightening. Some statistics could be better integrated into a cohesive narrative. The transition between global and Ethiopian data could also be smoother. There are slight redundancies in defining hypertension and discussing its impact, especially in the first paragraph.

5. Introduction (Lines 104–150):

The reason why you conducted this study was ‘’the review of previous Ethiopian studies mentions inconsistencies.’’ Could you elaborate on what specific factors showed differences across studies?

The second main reason you mentioned ‘’No other study conducted on this title in this study area? This is not enough reason to you. Because as I mentioned in question 3 above many studies conducted in Ethiopia. Remove or modify.

6. Study Questions (Lines 155–159)

Comment: The study questions are clearly stated. However, the second question could be expanded to specify whether the focus is on modifiable factors.

7. Methods - Study Design (Lines 169–230)

Comment: The study design is well explained, but the sample selection and sampling methods could be clarified further with diagrams or tables for better understanding.

The inclusion and exclusion criteria are reasonable, but it would be helpful to briefly mention why pregnant women were excluded (e.g., due to potential pregnancy-related blood pressure changes).

8.Sample Size Determination (Lines 182–202)

Comment. The sample size determination is well explained, but the rationale for choosing specific power and proportion assumptions should be clarified.

Suggestion: Consider explaining whether a sensitivity analysis was performed to determine if a different sample size would yield different results.

9. Sampling Procedure (Lines 182–205)

Could you clarify how the study handled patients who missed their follow-up visits during the study period?

10.Data Collection Tools (Lines 205–238)

The data collection process is described in detail, but more explanation is needed on why specific scales like H-SCALE and HELM were chosen.

The decision to modify the scales should be elaborated to help readers understand the implications of these changes.

11. Operational Definitions (Lines 252–295)

The operational definitions are clear, but certain concepts, such as adherence to medication and diet, could be expanded further to clarify how they were measured in real-time versus self-reported adherence.

12. Data Quality Management (Lines 314–332):

Question: The steps to ensure data quality are mentioned, but there could be more emphasis on minimizing bias. Was inter-rater reliability tested for the nurses collecting the data?

13. Results (Lines 345–385)

Comment: The results are presented clearly. However, the discussion on BMI and age could benefit from deeper contextualization. The high proportion of patients not adhering to medication or diet (51.2% and 79.4%, respectively) warrants further discussion on the reasons for such low adherence.

-Suggestion: Including more visual elements (e.g., charts or figures) would make the results easier to digest.

14.Discussion (Lines 399–492)

Comment: The discussion compares the results to several international studies, which is good. However, it would benefit from mentioning how cultural factors, healthcare access, and systemic differences might explain the disparities between Addis Ababa and other global cities.

- It would also be helpful to explore more deeply why certain factors (e.g., diet non-adherence or the number of antihypertensive medications) are so prevalent in this population.

15. Limitations (Lines 495–502):

The study limitations are acknowledged, but they could be expanded. How significant do you think social desirability or recall bias was?

16. Conclusion (Lines 503–517)

The conclusion summarizes the findings well, but it could be more impactful by highlighting the policy implications. The call for "strategies for food vendors to prepare low-salt diets" could be elaborated with more practical recommendations.

Good luck!

6. PLOS authors have the option to publish the peer review history of their article (what does this mean?). If published, this will include your full peer review and any attached files.

Reviewer #1: No

Reviewer #2: **Yes: **Temesgen Anjulo Ageru

---

## [Author Response · Author response to Decision Letter 0]

17 Oct 2024

Above all, we thank you for your constructive comments and helpful suggestions that helped us to improve and enrich our study

---

## [Decision Letter · Decision Letter 1]

5 Nov 2024

PONE-D-24-37212R1Uncontrolled Hypertension in Adult Patients in Addis Ababa Public Hospitals: Prevalence and Associated FactorsPLOS ONE

Dear Dr. Worku,

Thank you for submitting your manuscript to PLOS ONE. After careful consideration, we feel that it has merit but does not fully meet PLOS ONE’s publication criteria as it currently stands. Therefore, we invite you to submit a revised version of the manuscript that addresses the points raised during the review process.

Thank you for addressing the initial comments provided by the reviewers. Based on the revised manuscript, I suggest you consider my comments and suggestions before the manuscript can be accepted for publication. 

We look forward to receiving your revised manuscript.

Kind regards,

Muhammad Haroon Stanikzai

Academic Editor

PLOS ONE

Additional Editor Comments :

Thank you for addressing the initial comments provided by the reviewers. Based on the revised manuscript, I suggest some further changes before it can be accepted for publication in this prestigious journal.

1. It would be preferable to change your title " Uncontrolled Hypertension among Adult Hypertensive Patients in Addis Ababa Public Hospitals: A cross-sectional study of Prevalence and Associated Factors"

2. Please rephrase lines 33-34.

3. Rephrase and summarize with two pages from lines 47-154.

4. Line 57: Please remove dot before hypertension.

5. Line 58: Please update reference 3 and 4 with the recent Lancet publications on global burden of 288 causes of death and Burden of 88 risk factors.

6. Please use the complete form once followed by the abbreviation. You have written the complete form of LMICs three times (Lines 67, 87,..).

7. Please delete study questions. You have already mentioned them in lines 151 and 152.

8. Please make Figure 1 clearer. Please remove the green color.

9. In operational definitions; Provide details what was the definition for medical comorbidities in the study.

10. Figure 2. Please delete figure legend from inside the figure and also follow PLOS ONE's figure requirements.

11. Figure 3. Please delete figure legend from inside the figure. The pie chart can be further improved. Please look at the paper for reference https://bmcpublichealth.biomedcentral.com/articles/10.1186/s12889-020-09838-4.

12. Figure 4. Please delete figure legend from inside the figure. The pie chart can be further improved. Please look at the paper for reference https://bmcpublichealth.biomedcentral.com/articles/10.1186/s12889-020-09838-4.

13. The conclusion section is redundant, and could be condensed to focus on the key findings and implications (not more than a half page).

14. The language and the flow of the writing should be improved.

Reviewers' comments:

Reviewer's Responses to Questions

**Comments to the Author**

1. If the authors have adequately addressed your comments raised in a previous round of review and you feel that this manuscript is now acceptable for publication, you may indicate that here to bypass the “Comments to the Author” section, enter your conflict of interest statement in the “Confidential to Editor” section, and submit your "Accept" recommendation.

Reviewer #1: All comments have been addressed

Reviewer #2: All comments have been addressed

2. Is the manuscript technically sound, and do the data support the conclusions?

Reviewer #1: Yes

Reviewer #2: Yes

3. Has the statistical analysis been performed appropriately and rigorously? 

Reviewer #1: Yes

Reviewer #2: Yes

4. Have the authors made all data underlying the findings in their manuscript fully available?

Reviewer #1: Yes

Reviewer #2: Yes

5. Is the manuscript presented in an intelligible fashion and written in standard English?

Reviewer #1: Yes

Reviewer #2: Yes

6. Review Comments to the Author

Reviewer #1: Thank you for the opportunity to review this revised manuscript titled "Uncontrolled Hypertension in Adult Patients in Addis Ababa Public Hospitals: Prevalence and Associated Factors." I appreciate the authors’ responsiveness and diligence in addressing all previous comments.

The authors have thoroughly addressed each suggestion, resulting in a clearer and more comprehensive presentation of the study. I am pleased with the revisions, which have strengthened the manuscript, and I have some of comments to considerate below.(as attached)

Reviewer #2: Thank you very much for your valuable correction made for my previous questions, comments and suggestions. I have no further comments and suggestion.

Good luck!

7. PLOS authors have the option to publish the peer review history of their article (what does this mean?). If published, this will include your full peer review and any attached files.

Reviewer #1: No

Reviewer #2: **Yes: **Temesgen Anjulo Ageru

---

## [Author Response · Author response to Decision Letter 1]

30 Nov 2024

For all reviewers and editor, we thank you for your constructive suggestions and comments on this manuscript that would improve the substance and content of the study. The authors are thankful for all contributors in this manuscript.

---

## [Editor Report · Decision Letter 2]

4 Dec 2024

PONE-D-24-37212R2Uncontrolled Hypertension among Adult Hypertensive Patients in Addis Ababa Public Hospitals: A cross-sectional study of Prevalence and Associated FactorsPLOS ONE

Dear Dr. Worku,

Thank you for submitting your manuscript to PLOS ONE. After careful consideration, we feel that it has merit but does not fully meet PLOS ONE’s publication criteria as it currently stands. Therefore, we invite you to submit a revised version of the manuscript that addresses the points raised during the review process.

There are still points that need to be considered before this manuscript can be accepted for publication.==============================

We look forward to receiving your revised manuscript.

Kind regards,

Muhammad Haroon Stanikzai

Academic Editor

PLOS ONE

Journal Requirements:

Additional Editor Comments:

- Uncontrolled hypertension: How was outcome defined? What about patients who do not have DM and chronic kidney diseases but they were taking AHMs.

- I do not understand what the authors are explaining from lines 245-266. The author should only provide details for their method of measuring adherence.

- Please delete figures 2 and 3. These data are available in Tables.

- Please revise in-text citations. Please use []. Moreover, please merge citations when you are using more than one citation.

- The font in Tables and text is different.

- The article needs to be corrected for language and grammar. There are still may capitalization and punctuation errors.

---

## [Author Response · Author response to Decision Letter 2]

10 Dec 2024

Above all, We thank you for your constructive suggestions and comments on this manuscript that would improve the substance and content of the study

---

## [Editor Report · Decision Letter 3]

11 Dec 2024

Uncontrolled Hypertension among Adult Hypertensive Patients in Addis Ababa Public Hospitals: A cross-sectional study of Prevalence and Associated Factors

PONE-D-24-37212R3

Dear Dr. Worku,

We’re pleased to inform you that your manuscript has been judged scientifically suitable for publication and will be formally accepted for publication once it meets all outstanding technical requirements.

Kind regards,

Muhammad Haroon Stanikzai

Academic Editor

PLOS ONE

Additional Editor Comments (optional):

Thank you for addressing the reviewer comments.
---

## [Editor Report · Acceptance letter]

16 Dec 2024

PONE-D-24-37212R3 

PLOS ONE

Dear Dr. Worku, 

I'm pleased to inform you that your manuscript has been deemed suitable for publication in PLOS ONE. Congratulations! Your manuscript is now being handed over to our production team.

Kind regards, 

on behalf of

Dr. Muhammad Haroon Stanikzai 

Academic Editor

PLOS ONE